# Gastronomic Experience and Consumer Behavior: Analyzing the Influence on Destination Image

**DOI:** 10.3390/foods12020315

**Published:** 2023-01-09

**Authors:** Alina Kovalenko, Álvaro Dias, Leandro Pereira, Ana Simões

**Affiliations:** 1DMOGG, ISCTE-IUL, 1649-026 Lisbon, Portugal; 2BRU-Business Research Unit and DMOGG, ISCTE-IUL, 1649-026 Lisbon, Portugal

**Keywords:** gastronomy experiences, satisfaction, destination brand image, food tourism

## Abstract

Gastronomy experiences are becoming a fundamental factor that influences the making of a decision regarding choosing a travel destination, as well as being a crucial factor in shaping tourists’ satisfaction regarding their overall travel experience. The aim of the study is to identify and explain the simultaneous impact of the key factors that influence a gastronomic experience and their impact on tourists’ satisfaction with a trip and the destination’s brand. These issues were addressed within the context of Ukraine, as this is an overlooked area of academic research, and an online survey was conducted, targeting domestic and international tourists. Structural equation modeling was used to assess and reveal the proposed hypotheses in the model. The study contributed to the theoretical understanding of the key factors that increases the occurrence of a memorable gastronomic experience and the relationship between the experience of food and its role in the satisfaction of and the perceived brand of a destination. Moreover, the finding showed that past experience and prior knowledge have a positive influence on the gastronomy experience, while tourists’ prior knowledge effects the perceived quality of a destination’s cuisine, as well as the food activities in the destination. Linkages in the model were empirically supported by statistical analyses. Nonetheless, the various level of the tourists’ involvement with gastronomy might be used as an input to examine and improve the memorable gastronomic experience on-site. The research simultaneously highlighted the importance of gastronomy to tourist destinations for positioning on international and domestic markets. The paper not only provides theoretical but also practical implications. The hospitality and tourism businesses benefit from acknowledging the importance of local food and the local food market. The findings of this study are also deemed to assist destination marketers who observe that tourists have become more demanding in search of unique experiences offered by destinations.

## 1. Introduction

Nowadays, the fast-growing tourism market has integrated food into its offerings to attract visitors and differentiate itself from other destinations by using unique culinary features and traditions [1,2]. Food has become a vital engine not only for increasing a destination’s attractiveness but also to add value to the place and provide economic benefits. Thus, gastronomy tourism is a way for a destination to prosper by showcasing their local cuisine cultures and local products and contribute to the destination’s brand image.

However, the study by Choe and Kim [3] confirmed that tourists are not interested in consuming ‘traditional’ tourism products, rather they seek to have new experiences that allow them to fully explore the cultures and traditions of the place. Due to the fact that, historically, destinations have been using food as an instrument for attracting tourists, the cuisine has gone beyond the daily routine and become a crucial element of the tourists’ experience [1]. As Bertella [4] pointed out, gastronomy tourism is such a type of tourism where tourists are on a quest for new experiences that are connected with local cuisine and allow them to experience the local culture. Gastronomy experiences are considered to be a way of participating in another culture through trying new recipes and ingredients, meeting locals, and visiting places with strong culinary identities [5]. Therefore, Hall et al. [6] indicated that gastronomy experiences are a window to the culture of the destination. Since consuming food applies to emotions and brings enjoyment, studies suggest that gastronomy experiences contribute to the pre-, during, and post-evaluation of the trip [3]. Consequently, studies suggest that culinary experiences contribute to shaping one’s personal satisfaction with the trip.

A large amount of discussion and debates have been conducted about the importance of creating memorable gastronomy experiences in the destination. There are studies which were done by Leong and Karim [7], Choe and Kim [3], and others that analyze the relationship between the satisfaction with a gastronomy experience and a revisit intention; food experiences and crafting a destination’s brand, restaurants, and destination promotion; and the motivation of tourists and a food trip. However, the complex approach to the food experiences encounters a wide range of influence factors and links with tourists’ overall satisfaction and a destination’s brand image remain relatively unexplored.

It is also worth mentioning that in parallel with the growth of tourism, the demand of a trip to places of war, military conflicts, and disasters have grown over the past few decades. Several researchers such as Causevic and Lynch [8], and Arnaud [9] have regarded tourism as a useful instrument for countries in the post-war period because it can help enhance their process of recovering. Despite the emerging streams of research in this field, there is a limited number of works into the links between gastronomy tourism and its role in post-war periods. More precisely, there is no research examining the gastronomy experiences in Ukraine. However, due to the ongoing war in the country, it is vital to analyze possible tourism tools which could be used during the post-war period to contribute towards the recovery of the national economy. There is no doubt that a battleground on Ukrainian territory brings devastating consequences to its heritage as well as an attempt to destroy the country’s national identity. From this point of view, highlighting the importance of Ukrainian gastronomy as a vital element of the heritage and integral component of the culture is needed. 

The study will contribute to the academic literature regarding the gastronomic experience of domestic and international tourists by showing the key factors that influence gastronomy experiences and clarifying the relationship between them. Thus, the current research aims through a review of the existing literature provide a comprehensive picture of food tourism in Ukraine and consider gastronomy experiences as a powerful tool during the post-war period. Another goal of the research is to establish the structural model that helps to explain the phenomenon of gastronomy experiences examining influencing factors as well as its impact on tourists’ satisfaction and a destination’s brand image, which are vital indicators for the tourism prosperity of the country [10]. Structural equation modeling (SEM) is used as the primary research method using SmartPLS 3 software. 

Hence, the current research will overview the existing literature regarding gastronomy tourism and gastronomy experiences, providing an overall picture of this phenomenon. The following section will describe the hypothesis’ development considering various items. Then, variables such as past experience, prior knowledge, cuisine, restaurants, the gastronomy experience, satisfaction, and a destination’s brand conflate into the conceptual model. Following this step, the methodology for the collection of the data and the obtained data will be analyzed. The next section is devoted to the discussion of the obtained results, where our hypotheses are tested in the integrated model. Theoretical and managerial implications, limitations, and recommendations for future research will be covered.

## 2. Literature Review and Conceptual Model

### 2.1. Key Concepts

In recent times, the usage of the term gastronomy tourism for the appellation of food-related types of tourism has increased [11]. Scientists and practitioners of the tourism sector also widely use definitions such as food tourism, gastro-tourism, wine tourism, gourmet tourism, and culinary tourism [12]. Over recent decades, tourism studies have generated a plethora of different definitions in order to describe this type of travel. For example, UNWTO stated that ‘gastronomic tourism applies to tourists and visitors who plan their trips partially or totally in order to taste the cuisine of the place or to carry out activities related to gastronomy’. Santich ([13], p. 20) described culinary tourism as ‘tourism or travel motivated, at least in part, by an interest in food and drink, eating and drinking’. Long [14] defined gastronomy tourism as a participation in food-related activities mainly driven by the desire to discover and experience local culture and traditions. All previously grounded definitions of gastronomy tourism are important as they cover different prospects of the phenomenon. It is worth mentioning that gastrotourism includes trips where eating and drinking is not the main aim of the trip but rather a part of the whole trave journey. Moreover, from Long [14]’s point of view, gastro-trips in the modern world are a way to explore the traditions of the community and deeply understand the local culture. Thus, from our point of view, food tourism should be considered primarily as an experience which allows, through tasting local products or dishes, participating in the cooking prosses, visiting farms or other venues, or learning the traditional ways of food preparation, tourists to consider intangible heritage.

Broadly speaking, food trips may include visits to local farmers and producers, local food fairs and markets, gastronomy events and festivals, participation in cooking masterclasses, and the tasting of meals and beverages [15]. Moreover, gastronomy tourism provides a huge variety of tourism products and services such as wineries and vineyards, breweries, food trails, cuisine-related seminars, blogs and vlogs of a foodie, thematic magazines, books, and guides [16]. 

In contrast, McKercher et al. [17] argued that the majority of travelers consume food during their stay in their destinations (dining in restaurants, trying a variety of cuisine styles, and buying from local markets and producers), which is not obligatory, supposed to be separated into a different group of tourists called food seekers. McKercher and Chan [18] pointed out that gastronomy is not always a primary reason for travel, even if tourists take part in a wide variety of food-related activities. For this reason, in the current research, we take into consideration general tourists whose primary motivation for visiting Ukraine was not exceptionally culinary. Some studies have revealed that gastronomy emerges as a prominent part of tourists’ lived experiences despite the fact of their country of origin, destination choice, cuisine preferences, and decision-making prosses [19,20].

However, Timothy and Rone [21] brought light to the concept that gastronomy tourism and the food itself present are parts of the broader cultural system. They pointed out that local cuisine traditions are a silent but crucial signpost of destinations’ heritage that represents a truthful way of living in the community. In this vein, Richards [22] added that cuisine and gastronomy behavior represent the cultural practices of the destination. There is evidence that local cuisine is significantly influenced by traditions, beliefs, cultural values, history, and relationships with nature within a particular community [23]. The climate, environment, and resources have influenced the methods of preparing and preserving food. Consequently, cuisine is an expression of a community’s lifestyle and cultural and natural heritage. Moreover, in some destinations, depending on their historical evolution, people celebrate festivals related to food harvesting (Makoviaa in Ukraine) [24]. Another example of the influence on eating habits is Mongolian nomads whose patterns, attires, and celebrations were dependent on the availability of food. Thus, Hillel et al. [25] argued that cuisine is an integral part of a community’s identity and serves as a tool for understanding and appreciating the local culture.

Hence, food has switched from being a necessity of tourists’ trips to a significant element of the travel experience [26]. As Ogden et al. [27] noted, tourists tend to learn and discover a destination through the food of the place. Moreover, they consider the variety and quality of places where food could be consumed, which is an important factor contributing to their satisfaction. Hjalager and Richards [28] pointed out that local gastronomy as a tourist product remains a vital factor in making a destination choice by tourists.

’Experience has been considered an intangible process which provokes individual interaction with the process (reference). As a result, individuals focus and concentrate on the activity upon the influence of personal beliefs, senses, emotions, backgrounds, and context in what experience takes place’ [29]. Mohamed et al. ([30], p. 33) defined experience as:
‘The total outcome involving a combination of customer’s cognitive, affective, emotional, social, and physical responses gained from participating in activities and interacting with both tangible and intangible components in the consumption process, which in turn influences how consumers interpret the world.’


Pine and Gilmore [31] highlighted that experience is carried out as an impression shaped by a personal engagement with products and services. They also noted that it includes a concentration of sensorial values in the realm of entertainment, education, escape, and aesthetics. Moreover, Youngman and Hadzikadic [32] suggested that an overall experience is a very wide and complex phenomenon as it compresses a variety of sub-experiences which happen within different conditions and on different levels. From these prospects, tourism is an example of a social complex occurrence which provides society with a range of experiences [33]. Notably, the latest studies confirm that gastronomy experiences have become the main element of the overall tourism experience [34,35].

Over the past years, tourists have shifted their way of travel from observing, learning, and exploring to experiential and fully transformative trips [8]. Tourism researchers have determined three stages of the evolution of gastronomy experiences [16,22]. The first generation of food experiences is focused on creating themed experiences with provoking sensory attributes. The second stage is characterized as co-creation experiences that involve tourists’ direct participation. The third stage embarks on an era of integration of visitors in local communities, exchanging knowledge between tourists and locals, and creating a network centered on food and cuisine. As tourists’ demands are constantly changing, people nowadays seek to have authentic, culturally rich travel experiences. They are looking for co-creation and communication with local communities through a direct interaction in order to learn about their culture. 

Gastronomy experiences emerge from four main assets, which are intellectual, affective, behavioral, sensory, and effective [36]. More precisely, the intellectual aspect allows people to expand their knowledge about the destination by learning and gaining local-based information. Food experiences provide the opportunity of communicating and interacting with family or friends once they take part in common food activities [14]. Moreover, food experiences make people feel pleasure and enjoyment. Last but not least, gastronomy provides tourists with a chance to have a sensorial experience via tasting local cuisine [37]. 

There is no doubt that gastronomy tourism is perceived as an effective path to exploring and experiencing a destination in an authentic way. Thus, gastronomy features as an enabler of tourist cultural experiences [38]. Moreover, Hjalager and Richards [28] found that gastronomy experiences supply tourists with an atmosphere that contributes to creating memorable experiences. It was also stated that food experiences enhance the value of the destination as they represent the authenticity and identity of the local community. Considering the fact that gastronomy experiences involve visitors in the process of the production or consumption of dishes that might be indicated by tourists’ prior knowledge, such a type of experience eagers a direct interaction with the community’s ecosystem. As a gateway, tourists receive an immersive experience enhancing their further overall satisfaction with the trip [37,39].

### 2.2. Hypotheses Development

In tourism research, past experience has been treated as a factor that directly influences tourists’ motivation, perception, evaluation of the trip, and their overall decision-making activity. It has been discovered that after tourists visit a destination, they rely on their prior experience as their primary source of information. As a result, the cognitive effort is reduced, and the process becomes more natural and harmonic [40]. Visitors who have a past experience with the destination are able to work with accurate information and filter that which is irrelevant [41]. The study conducted by Tse and Crotts [42] confirmed that tourists who have experience with the destination significantly correlated with a variety of culinary explorations. Ryu and Jang [43] identified that past experience is a vital factor in the prediction of the tourist’s food consumption and validating the destination’s cuisine quality. Moreover, accumulated experience is perceived as a strong factor that influences not only tourists’ expectations and perceptions but also their evaluation of their future experience [19]. 

As has been already stated, the degree of tourists’ personal affection for tourism activity is continually underpinned by engagement, knowledge, and previous experience [15]. For example, a tourist who is interested in gastronomy and food-related activities usually has more experience with such activities. This in turn strengthens gastronomic knowledge. From this point of view, past experience improves knowledge, which is a cognitive reaction to the gastronomy ingested. Thus, it is assumed that a tourist’s cognition would produce a stronger perception of the place, its cuisine, activities, and gastronomy level [43]. Moreover, from the prospects of gastronomy tourism, travelers with strong preferences for the destination’s cuisine tend to have more intended recurrent gastronomic experiences. Thus, the perception of food activities and the quality of local cuisine and restaurants are stronger. 

It has been found that past experience is a vital factor that influences customers’ post-evaluation. This is important to take into consideration while exploring a customer’s perception of visiting restaurants. Thus, because of the influence of this variable on customers’ response to the consumed product, first-time consumers do not perceive the service and product in the same way as repeated tourists. It is noteworthy that guests who have previous experience of visiting the restaurant tend to repeat patronage because of a clearer understanding of what to expect [44]. Moreover, repeated visitors tend to have a better perception and higher satisfaction with dining in a familiar place rather than first-time guests. 

According to the literature review, the research hypotheses to be tested are as follows: 

**H1a.** 
*Past experience positively influences the perception of the local cuisine quality.*


**H1b.** 
*Past experience positively influences the perception of the quality of local food activities.*


**H1c.** 
*Past experience positively influences the perception of the gastronomic experiences’ quality.*


**H1d.** 
*Past experience positively influences the perception of the local restaurants’ quality.*


It is recommended by the academic literature that knowledge regarding the destination could be divided into informal and formal groups [45]. Thus, in the first case, tourists receive information about the place and attraction from relatives, friends, bloggers, or other people who have previous experience with visiting the destination, which is referred to as ‘word of mouth’. However, formal knowledge refers to those sources such as travel guidebooks, journals, other types of printed materials, and online types of advertisements [46]. Typically, the information obtained from these sources includes details of the cuisine, gastronomy traditions, and activities of the place as well. Thus, those tourists’ knowledge of local gastronomy is closely connected to the dining past experience as well as their knowledge which has been gained formally and informally [7]. 

In the field of tourism research, empirical studies have confirmed that knowledge about local cuisine could be a factor that motivates gastronomy-interested tourists to be active partakers in local food activities and gastronomic experiences despite the general interest in local food attributes among average tourists. Nonetheless, it has been confirmed that regardless of whether tourists visit the destination for gastro purposes or if food is just a part of the overall experience, the presence of knowledge about gastronomy has a positive effect on tourists’ satisfaction with gastronomy-related activities and the cuisine’s quality [15]. Thus, it has a positive influence on the overall perception of the gastronomy of the destination. Moreover, previous studies pointed out that the formation and acquisition of knowledge regarding the gastronomy of the place is a vital cognitive trigger in a tourist’s local food taste perception journey. Hence, the knowledge received from the experience of consuming local food through visiting restaurants positively influences the overall perception of local restaurants’ quality [47]. Thus, based on the review of previous studies, the following proposed hypotheses were developed:

**H2a.** 
*Prior knowledge positively influences the perception of the local cuisine’s quality.*


**H2b.** 
*Prior knowledge positively influences the perception of the local food activities’ quality.*


**H2c.** 
*Prior knowledge positively influences the perception of the gastronomic experiences’ quality.*


**H2d.** 
*Prior knowledge positively influences the perception of the local restaurants’ quality.*


Concerning the dimensions of food activities, food activities play a vital role in building destination brand image as well as show a positive correlation with the tourists’ satisfaction [7]. Food activities are perceived as an integral part of the tourist travel experience and could be presented by food-tourism festivals, the wine and food offer of various regions, street food markets, gastronomy-related tours, cooking master classes, visiting local farms or local food manufacturers, etc. [7]. Food festivals and events as the main contributor to the brand image of the destination and a vivid factor of tourists’ satisfaction remain the most investigated topic by scholars. However, there is a limited number of studies where there is an identified direct link between festivals and their role in the formation of a destination’s brand image. 

Yang [48] looked at the effect of the Taiwanese coffee festival on tourists’ overall satisfaction with the destination. Lee et al. [49] has paid attention to the relationship between the guests’ satisfaction, emotions, and the tourist revisit intention, where they argued that the affective and cognitive satisfaction induced by the festival has a further impact on a tourist’s attitude to the place and their level of satisfaction. Robinson and Cliffor [50] suggested that service escape, hygiene rules, and authenticity are the key determinants of tourists’ satisfaction from an event and destination, which they then put into perspective. 

Bjork and Kauppinen-Raisanen [44] have tested the influence of food masterclasses and tours in Finland. The main findings of the study confirmed that the variety, authenticity, and quality of the food activities have an extremely strong impact on tourists’ holiday satisfaction within particular tourist groups. For example, France and Italy have built their image as premier food destinations and established themselves as the ‘land of wine’ by using a variety of food activity offers [7]. Hence, with regard to the findings on the previous studies, the following hypotheses for further testing were developed: 

**H3a.** 
*The perception of the local food activities’ quality positively influences the destination brand image.*


**H3b.** 
*The perception of the local food activities’ quality positively influences the tourists’ satisfaction.*


Another vital attribute of destination gastronomy is restaurants where the majority of local food and culinary culture tends to be consumed. Those places are the representation of the local culinary identity and offer a variety of dining experiences, starting from gourmet restaurant which serves high cuisine, local casual cafes offering local meals, to street food markets and fast-food places [51].

There is a widespread assumption that a restaurant is one of the main sources of selling food. Although food is a focal point, in current times, restaurants provide ‘food service experiences’. The analysis of the existing literature has shown that a restaurant service is the combination of tangible and intangible elements [52]. In other words, there is a place where producing and consuming the experience of a service take place simultaneously. A defining moment occurs when the provider delivers a service to the customer and they are able to demonstrate its quality. Thus, these interactions have an impact on the further evaluation conducted by and the satisfaction of the visitors. Some previous studies have stated that guests evaluate their dining experience by focusing not only on the product itself but also by considering the way in which it was delivered functionally [53]. More precisely, the quality of the dishes might be spectacular, but if service is slow or the waiter is unqualified, the outcomes will dismiss the tourists’ overall satisfaction. Hence, the importance of functional quality prevails over the technical one. Research has found that pricing is a vital attribute for the evaluation of local restaurants, which also contributes to the judgement of their quality [43]. The majority of people tend to attend and return to a restaurant where they believe they are receiving value for money. This determinant varies from individual to individual.

Another aspect of a customer’s satisfaction contributed to by their dining experience was a distinguished service environment. It has been confirmed that the environment the service is delivered within has a substantial impact on the people’s reaction to the place, social interaction, and further behavior [44]. People tend to choose those places where the environment enhance their feeling of pleasure and fulfillment. Moreover, some scholars postulated that such conditions of service such as seating arrangements, dimensions, and flexibility influence the tourists’ overall evaluation of the restaurant [54]. Nevertheless, some studies revealed that the dining atmosphere and environment has a strong impact on a destination’s brand image. It is noteworthy that scholars emphasize that such attributes to the dining experience, such as the availability of multilingual (or at least in English) menus, highly qualified staff members, and various meal options, also contribute to the positive brand image of a destination [7].

Additionally, the perceived quality of the restaurant may affect the customers’ psychological, cognitive, and emotional responses, which in turn impacts their satisfaction with the restaurant as well as the whole gastronomy image of the destination. Based on the outcomes of the received service, it might also influence peoples’ beliefs and their perceived image of the place as the emotional response was elicited. According to the literature review, the research hypotheses to be tested are as follows: 

**H4a.** 
*The perception of the local restaurants’ quality positively influences the destination’s brand image.*


**H4b.** 
*The perception of the local restaurants’ quality positively influences the tourists’ satisfaction.*


Over the past decades, destinations have begun to have the desire to differentiate themselves from others and promote their own cultural identity. Destination tourism experts focus on developing unique experiences which turn to creating positive and recognizable brand images [55]. In the early tourism literature, Cromton ([56], p. 18) defined a brand image as ‘the sum of beliefs, ideas and impressions that a person has of a destination’. Sun et al. [57] added that a destination’s brand image is a tourist’s own elaboration and interpretation of their knowledge, beliefs, assumptions, emotional impressions, and ideas regarding a place. In recent years, scientists and practitioners of tourist research have started to consider local cuisine and gastronomy as one the key elements of successful destination branding. Following this concept, Berg and Sevon ([58], p. 289) postulated that ‘food and gastronomy is directly and indirectly affecting the character of the place and its brand-image’. In this vein, the World Tourism Organization highlighted that ‘tourists travel to those destinations that have established a reputation as a place to experiment with quality local products’. Several studies have supported the idea that quality is important not only to provide a memorable experience but also to create the distinctive brand image of a destination [59]. Additionally, Beerli and Martin [60] argued that the gastronomic experience is 1 of the key determinants within 24 elements that contributes to the overall brand of a destination. Once authentic and unique features are implemented in the food experience in the destination, a positive gastronomy image is crafted which enhances the overall branding of a destination [61].

Another vital prospect aspect of the tourists’ trip is their level of satisfaction. In the competitive economy, it is important to understand how tourists evaluate their satisfaction with a trip and what factors determine it. Thus, Lopez-Guzman and Sanchez-Canizares [62] argued that modern tourists are no longer satisfied just by a product itself, they seek to have an authentic cultural experience and discover a destination’s traditions. In this sense, Martín et al. [63] argues that cuisine is a pivotal attribute of a destination’s cultural heritage as the richness of the local products and recipes represent a destination’s identity. Hence, scholars have pointed out that the gastronomy experience is an exceptional source of a tourist’s satisfaction [64]. From this perspective, Medina-Viruel et al. [65] supported that somebody’s level of satisfaction varies and is related to the food experience they receive in the visited destination. According to the results obtained by Guan and Jones [46,66], travelers who highly rate the quality of their gastronomy experience tend to express a higher degree of satisfaction with their whole trip. In accordance with the scientific literature review, the research hypotheses are as follows:

**H5a.** 
*The perception of the gastronomic experiences’ quality positively influences the destination’s brand image.*


**H5b.** 
*The perception of the gastronomic experiences’ quality positively influences the tourists’ satisfaction.*


Regarding the building of a strong brand for a destination, scholars always emphasize the necessity of creating a successful brand image, which is the perception of a brand in terms of its qualities and associations, usually organized by tourists in a meaningful way and held in their memory [67]. Thus, a powerful and recognizable destination image immediately contributes to creating an overall brand and enhancing it as well. Nevertheless, food and cuisine have been distinguished as the crucial elements in a destination’s branding. Research states that while destinations tend to build a strong brand focusing on gastronomy tourism, the quality of the food and cuisine became its selling point within tourism branding strategies and national tourism boards [68]. Several studies confirmed that countries or particular destinations which aim to increase their competitiveness enhance their brand by recognizable and favorable food and cuisine attributes [69,70].

Some scholars state that quality is one of the key elements that determine and shape the tourists’ overall satisfaction. The quality of the local cuisine might be explained by a variety of characteristics of excellence which make the local food sufficient for visitors, for instance, the taste, freshness, nutritive value, safety, and type of ingredients. Thus, researchers such Meng et al. [71] and Coreira [72], and others have confirmed in their studies that the perception of local cuisine is a pivotal element for tourists’ satisfaction. Aiming to better understand the term quality of food, they consider it from different aspects. Hendijani [73] discovered that using fresh and healthy ingredients in local dishes, as well as their taste, contributes to the overall quality of the proposed food. 

Moreover, Seo et al. [74] confirmed that being able to taste healthy and safe food during their trip increases the tourists’ intentions to discover local cuisine and, respectively, enhances their satisfaction. Sthapit [1] found that the quality of local food encounters is determined by the characteristics of being novelty, generous, and delicious. Furthermore, food hygiene was found to be an important aspect in the context of the tourists’ satisfaction with the local cuisine [75]. Furthermore, the variety of dishes, the possibility of different meal combinations, as well as the price are factors which are considered to be important attributes for assessing the quality of local food [72]. Thus, in accordance with the scientific literature review, the hypotheses to verify shall be as follows:

**H6a.** 
*The perception of the local cuisine’s quality positively influences the destination’s brand image.*


**H6b.** 
*The perception of the gastronomic local cuisine’s quality positively influences the tourists’ satisfaction.*


## 3. Materials and Methods

This section addresses the explanation of the methodology that has been used in this research paper. It introduces the research philosophy of the paper and how it was applied. The section also goes on to show the method of the data collection and presents the description of all of the measurements of the survey. It explains the data analysis strategy and the tools that were involved in the research prosses. Moreover, the section reveals the sample features as well as emphasizes the context of the research area to provide a better understanding of the context. 

The research was conducted with a positivist and deductive approach, with the application of quantitative techniques. The positivist approach has been used in order to verify a prior hypothesis by showing the relationship between the casual and explanatory factors (independent variables), and the outcomes (dependent variables) [76]. As research attempts to explain the complexity of the gastronomy experience phenomenon, the positivist approach has been selected due to its capability to show a correlation among a variety of variables and verify the influence of the explanatory factors as well as the outcomes [77]. Thus, the positivist approach facilitates the explanation and prediction of tourist behavior regarding gastronomy experiences in Ukraine and its influence on a destination’s brand.

### 3.1. Measures

In order to collect data, an internet-based questionnaire was used. The questionnaire was created through the literature review by implementing the measurement scales for each item from different research that was based on the evaluation of items via the Likert scale [7,78]. To ensure the reliability and validity of the questionnaire, the questions, constructs, and items were based on relevant previous research and were adjusted to the validity of the context (see Table 1). 

The internet questionnaire was primarily disseminated by social media sites such as Instagram, Facebook, Telegram, and emails between December 2021 and February 2022. Those channels of distribution for the questionnaire were chosen as based on the report of the State Agency of Tourism Development [81,82] in Ukraine where it was stated that such social medias are considered to be the most popular and the most used sources to search information, make bookings, communicate, and for other tourism-related purposes while traveling.

All participants who have joined the survey took part voluntary and participated based on informed consent as they were previously advised about the purpose, benefits, and risks and were provided with sufficient information about the future implication of the results. The collected data do not include personally identifiable information, so the questionnaire is completely anonymous. The data collection and analysis process follows confidentially principles and respects the participants’ privacy. The paper retains the objectivity of the data analysis, discussion, and interpretation of the results; it also acknowledges the use of the academic works of other authors following the APA referencing system.

The questionnaire was created and divided into six sections. Thus, in the first part, the respondents were asked to fill out their demographic profile indicating their age, gender, profession, country, education level, and the existence of a gastronomic experience during their visit to Ukraine. The next parts were called to evaluate their gastronomic experience within the context of the trip. The second segment consists of the questions related to the food-image dimensions aiming to measure the various aspects of the experience, such as the perception of Ukrainian cuisine, the restaurant sector, and food-related tourism activities. A five-point Likert scale (1—strongly disagree to 7—strongly agree) was used. The third part of the survey encompasses questions regarding the tourists’ behavior and asked them to indicate their gastronomic experience as well as clarify their level of knowledge about local foods and evaluate a prior gastro experience. In this part, a five-point Likert scale (1—not important to 7—very important) was used to describe a gastronomic experience in Ukraine, while a five-point Likert scale (1—strongly disagree to 7—strongly agree) was used for the rest of the constructs. The fourth section includes questions that belong to a perception of the overall destination brand where respondents via the previously mentioned five-point Likert scale showed their perception of the Ukrainian brand as a tourist destination. The last part of the questions was created to ask people to evaluate the participants’ satisfaction with their trip to Ukraine.

SmartPLS 3 software [83] was used as a tool to estimate the proposed conceptual model by means of partial least squares structural equation modelling (PLS-SEM). Henseler et al. [84] argued that PLS-SEM evaluates the measurement model, which reviews the constructs’ validity and reliability, as well as the structural model, which examines the hypothetical relationships between the independent and dependent items. Moreover, one of the advantages of using this method lies in its minimal requirement for the size of the sample and residual distribution.

Thus, this algorithm is effective for studies where research is prediction-oriented or deals with complex models [85]. The validity and reliability of the structural model were examined by various tests. Speaking more precisely, applying to suggestions of Hair, Hult, Ringle, and Sarstedt [86], during the testing, the reliability, convergent validity, internal consistency reliability, and discriminant validity were examined. To provide an additional confirmation about the quality of the model, we tested for convergent validity by analyzing the standardized factor loadings and by estimating the average variance extracted (AVE). To test the discriminant validity, the Fornell and Larcker criterion was used [87] and complemented with the heterotrait–monotrait ratio (HTMT) criterion [86].

Regarding the predictive relevance of the conceptual model, it was assessed through the sign, magnitude, and significance of the path coefficients; the values of R^2^ and Q^2^ for each construct endogenous variable as a measure of the model’s predictive accuracy; and the Stone–Geisser’s Q^2^ [86].

To provide more in-depth insights about the results, an importance performance map analysis (IPMA) was conducted. Hair et al. [88] stated that IPMA is based on standardized regression coefficients (importance) and adds an additional dimension to the analysis that considers the values of the predictor variables. In other words, IPMA is used in order to extend the PLS-SEM results considering the performance of each construct. In this vein, IPMA allows us to drown the results by considering both dimensions, such as the importance and performance, which is important for making managerial decisions. Thus, practitioners have an opportunity to prioritize their action considering the obtained results of the IPMA.

### 3.2. Sample

The targeted population was tourists who have previously visited Ukraine at least once. The chosen approach has allowed us to receive 187 complete answers, which is an adequate number to test the proposed hypothesis and the model. Due to the fact that the total number of the members of the population is indetermined, establishing a sampling frame is impossible. As such, we followed a non-probability sampling procedure; more specifically, a purposive sampling technique was adopted. The snowball technique was implemented by asking respondents to spread the link for the question form. Involved virtual snowball [89] as the data collection strategy was chosen as the most suitable way to target the hard-to-reach population [90] since answers from international and domestic tourists who visited Ukraine are not easily accessible. It is also important to mention that snowball sampling is subject to various biases [90]. Thus, it is quite important to verify an accurate community and target the appropriate group from the beginning as the first participants who will be asked to encourage others to take part in the survey have the strongest impact on the overall sample and the future results. Although this is a limitation of this technique, it is considered appropriate in tourism studies to meet relevant participants in a research population [91].

The profile of respondents is described in Table 2. The sociodemographic profile was, as a majority, female (76.4%), the respondents were mostly in the age groups of 18–29 (76.4%) and 30–39 (20%), and either had a full-time part-time job (45.5%) or were students (36.4%). This sample is influenced by a high proportion of young people, who tend to have less money to spend in gastronomy. Taking into consideration the Ukrainian target market, which is primarily people aged 20–35 years who aim to partake in leisure activities and middle aged (35–54 years) travelers who visit the country for business purposes, the sample of the research covers those segments [82]. Furthermore, gastronomy is part of the tourist travelling experience, and visitors will spend money on food independently of their income [44]. Regarding their country of origin, most of the respondents are Ukrainians (56.4%), but 43.6% are international visitors from Poland, Romania, Germany, France, the USA, Italy, Portugal, and Turkey. This sample considers domestic tourists as part of the sample as other studies in gastronomic tourism [44]. According to the collected answers, 95% of people stated that they had a gastronomic experience while visiting the country. Additionally, the respondents’ profile presents a similarity to the demographic characteristic of the respondents from other research where gastronomy tourism and food experiences were studied concerning the fact that the sample mostly presented young people with a high education and a stable income [92].

### 3.3. Study Context

Ukraine has a great potential for the development of gastronomic tourism [82]. This is facilitated by the multinational nature of the country. Nowadays, Ukraine’s population consists of communities with a variety of different roots. Ukrainians and the representatives of other nations and nationalities have formed a unique ethnocultural face, an integral feature of which are unique culinary traditions. National Ukrainian cuisine is recognized in many countries around the world and attracts foreign tourists. Gastronomic tours include an introduction to traditional cuisine and a discovery of the peculiarities of the food customs [93].

According to the State Statistical Service, 57,712 restaurants were registered in Ukraine at the end of 2019. The largest number of them are located in the Kyiv region and the smallest are in the Ternopil region; they are equal to 4505 and 769 objects, respectively. Thus, we can observe that Kyiv, Odesa, Kharkiv, and Lviv are the regions where the number of food establishments exceeds the average in Ukraine. This is primarily due to the highest number of tourists flows as well as the rate of the region’s attractiveness. Lviv, Odesa, and Kyiv oblasts are the leaders in the regions where tourists visit in Ukraine. In addition, an important factor is the population of the administrative centers of the regions, because this is concentrated on the main number of restaurants [94].

Gastronomy tourism in Ukraine have been the focus of researchers such as Braichenko et al. [93], and others over recent years who considered food tourism as a perspective industry for the national economy, with a high potential to grow internationally. Moreover, the concept of gastronomic tourism is being actively developed in the form of a road map; the document will be presented and included in the national strategy for the development of tourism and resorts until 2026. According to the National Tourism Organization of Ukraine, 2018 was named the Year of Gastrotourism [82]. Experts and restaurateurs agree that gastronomic tourism is one of the most promising types of tourism in Ukraine, with which the country can be represented on the world market. According to the research, tourist flows are increasing in those regions where thematic fairs, tours, public holidays, and festivals are held. Many different gastronomic tours take place in Ukraine, which creates preconditions for the formation of a competitive tourist product in the domestic and foreign markets [82].

It should be noted that in Ukraine today, Borscht festivals are in demand among tourists, and the most popular is held in the village of Borshchiv in the Ternopil region. Every year, the fat festival traditionally gathers tourists in Petrykivka in the Dnipropetrovsk region. Lviv is famous for its amazing museum–restaurant ‘Salo’; the Zakarpattya and Volyn regions are focused on for their different sausages festivals which are popular among domestic visitors. Polissya is famous for its potato harvests and dishes; the Deryn Festival is held annually in Korostyn in the Zhytomyr Region and a monument to this dish has been erected in the city. Transcarpathia is known for the Berlybas Banosh festival, which takes place in the village of Kostylivka, Rakhiv district. Poltava hosts an annual dumpling festival, and a monument to this venerable dish has even been erected. Every year a festival of national cuisine is held in Lutsk, the main dish of which is dumplings. Moreover, specific interest receives master classes on cooking regional cuisine, or short cooking courses, during which a person not only gains useful knowledge but also becomes acquainted with the national dishes and traditions [94].

More precisely, most popular today are trips in which offer experiences with cheese, wine, and honey tourism [94]. About 75 companies officially produce wine in Ukraine. The total area of vineyards is 40,700 hectares, of which 25,600 hectares are set aside for production of wine [93]. As a percentage of the wine production in Ukraine, the main wineries are located in the Odesa (55%), Zakarpattya (15%), and Kherson (15%) regions, which are the centers of wine tourism in Ukraine. Wine tours have received widespread popularity among domestic and international tourists. It covers many aspects: an acquaintance with the customs, traditions, and life of the region, meetings with winemakers, tastings of the best wine samples, and visits to family wineries and vineyards. For example, tourists have the possibility to enjoy the Ukrainian wine culture in the Wine Culture Center (Shabo village, Odesa region), take a tour through the ways of creating wine and taste Ukrainian Bessarabia, which includes a wide range of local food and wine producers, visit family wineries such as “Kurin” (Kherson), a winery of Prince PM Trubetskoho (Kherson region), Grande Vallee Winery.

Ukraine produces a wide variety of natural products, one of which is a variety of honey. The main number of apiaries is located in the east and south of the country. In total, honey is produced in 13 regions in Ukraine. The Mykolaiv area occupies the first place as it has several apiaries with the certificate of organic production; there are 23 apiaries here, which makes up approximately 7.5 thousand families of bees. There are also many snail farms in Ukraine that can provide the same services as the famous French ones. From 2014 to 2019, 200 snail farms were established in the country. They produce about 800 tons of live snails; in 2019 they exported 30 tons. These producers are in the Luhansk, Kyiv, Lviv, Donetsk, and Mykolaiv regions.

A necessary direction for the development of gastro tourism at the regional level and in the country as a whole is defining a gastronomic brand. The gastronomic brand of Ukraine enhances the overall image of the country as a producer or exporter of unique high-quality food products [95]. Branding is an important element for the development of a tourism destination, which is defined as the competitive identity of a place. The main components of the gastronomic brand are a well-developed field of gastronomy; the availability of specialists in the field of catering with the use of traditional products; the availability of authentic dishes (authentic products); and gastronomic events (festivals and competitions). A well-built gastronomic brand means that the food is not only enjoyed during the trip, but it also makes for delicious souvenirs that the tourists can bring home. Therefore, it makes the memories of the trip and forms the desire to cook at home according to the recipes he or she has learned during their visit, as well as a tendency to choose restaurants in this city.

## 4. Results and Discussion

Thus, with using the PLS-SEM, using Smart-PLS 3 software, we analyzed the collected data and tested the hypotheses. The results have disclosed that the standardized factor of the loadings of all of the items were significant (*p* < 0.001) and superior to 0.6 (the minimal value was 0.71). It provided us with the confirmation of the reliability of discrete indicators. Moreover, Table 3 and Figure 1 have presented results which indicate that the Cronbach alphas and composite reliability (CR) indicators are higher than 0.7 [86]. Consequently, it allows us to argue that the internal consistency reliability was confirmed for all constructs.

Another vital step was testing the convergent validity. As a result, taking into consideration that all the constructs’ items loaded positively and significantly, the convergent validity was confirmed. Furthermore, this was supported since the CR values were superior to 0.7 and the average variance extracted (AVE) surpassed the limit of 0.50 [96]. Consequently, the Fornell and Larcker criterion was adjusted to the study in order to test the discriminant validity. As could be observed from the diagonal shown in Table 3, the square root of the AVE of each construct is superior to its biggest correlation with any construct [87]. It is important to mention that the heterotrait–monotrait ratio (HTMT) criterion [84] was significant. Thus, we could argue from the obtained results in Table 3 that the discriminants are valid since the HTMT ratios are inferior to 0.85 [84,86].

There is a crucial step prior to the evaluation of the quality, which is the confirmation of the collinearity as it was advised by Hair et al. [86]. Thus, following the obtained results, the VIF values ranged from 1.18 to 4.45, meaning that they were inferior to the threshold of 5 [86], which, respectively, showed us the confirmation of no collinearity. Furthermore, we also assessed the R^2^ (the coefficient of the determination) for the six endogenous variables of Cuisine, Destination_brand, Food_activities, Gastro_experience, Restaurants, and Satisfaction which were 0.296, 0.474, 0.548, 0.690, 0.485, and 0.584, respectively, and superior to the limit of 10%. To obtain the Q^2^ values, a blindfold analysis was performed. The results for the endogenous variables (cuisine; destination brand; food activities; gastronomic experience; and restaurants satisfaction) were 0.155; 0.210; 0.228; 0.359; 0.193; and 0448, respectively. The values obtained were positive, providing additional support of the model’s quality.

To test the hypothesis, we conducted a bootstrapping analysis with 5000 subsamples to evaluate the significance of the parameter estimates [86].

The results presented in Table 4 show that the tourists’ past experience influences their perception regarding the quality of the cuisine (β = 0.290, *p* < 0.05). This result provides support for H1a. Thus, the findings coincide with the results from the previous research [15,43]. In another words, previous experience contributes to the creation of the memory of the food and influences the following perception and evaluation of the cuisine of the destination [97]. This finding is consistent with study of Leong et al. [7] and Ryu and Jang [43] where past experience has been confirmed to be a significant predictor of the visitors’ intention to consume the local cuisine.

However, the tourists’ past experience does not influence the perception of the quality of the food activities, thus H1b is not supported (β = 0.185, n.s.). So, it was discovered that within the context of Ukraine, past experience does not influence how people perceive the quality of the food activities and this factor is not vital in shaping the future of peoples’ evaluations and perceptions of the quality and variety of the activities offered in the destination. This result is in contrast with the study undertaken by Karim and Chi [98], who noted that a past experience could shape people’s attitude to the food activities offered in France, Italy, and Thailand, and the study conducted by Kala [53], who argued that enhancing and diversifying a previous experience helps to shape the tourists’ perception of the food activities. 

Although, it has been revealed that past experience has a direct impact on a gastronomic experience (β = 0.649, *p* < 0.05) and the perception of a destination’s restaurants’ quality (β = 0.595, *p* < 0.05), which confirms the proposed hypotheses H1d and H1c. Hence, the obtained results lie within the findings of previous studies [7,38]. The confirmations of those hypotheses allow us to argue that tourists who have had an experience which resulted in their intention to return have stronger preferences for the restaurants as well overall gastronomy, and hence this will better influence their perception of the quality of dining establishments and their general food experience. Meanwhile, with regard to previous research, the current study confirms that the quality of a past experience is a significant component which contributes to the perception of the overall gastronomy experience during the trip. Additionally, a familiarity with the place, specifically the destination’s restaurants, has a strong correlation with the general gastronomy experience.

Regarding prior knowledge, the results from the study have shown that it influences the perception of the cuisine’s quality (β = 0.325, *p* < 0.05), food activities (β = 0.620, *p* < 0.05), and the gastronomic experience (β = 0.268, *p* < 0.05). Hence, those findings provide support to hypotheses H2a, H2b, and H2c. Moreover, these results are consistent with previous studies [7,66], who verified the direct impact of prior knowledge on the perceived quality of the cuisine, food activities, and general gastronomic experience obtained in their destination of the choice. Hence, formal or informal knowledge received via different channels of communication has shown to have a significant impact on the tourists’ perception not only in terms of the quality of the local cuisine and food activities, but also in terms of their overall attitude towards the experience. As Kivela and Crotts [15] stated previously, our research has also supported the assumption that regardless of the type of tourists (if food was the main goal or just a part of the trip), knowledgeable tourists about cuisine have a higher satisfaction and, as a result, they would view their food experience more positively. Consequently, the quality, validity, and reliability of the knowledge about the gastronomy of the destination are important due to its strong relationship with perception of the gastronomy experience. Although, the assessment shows that prior knowledge do not influence the perception of a restaurant’s quality (β = 0.158, n.s.), so the hypothesis H2d was not confirmed. From this vein, the study has shown that the knowledge collected before the attendance to the restaurant will not influence its perceived quality. 

What is more interesting is that the results disclose that the food activities do not have an impact on the destination’s brand (β = 0.064, n.s.) and the peoples’ satisfaction of the place (β = 0.130, n.s.), thus hypotheses H3a and H2b were not supported. Even though some studies were partially in line with the proposed hypotheses [44,98] and showed a significant association between the food activities and the destination’s brand and satisfaction with it, our study did not confirm this assumption for tourists who have been to Ukraine.

Taking into the consideration restaurants, the analysis of the conducted data via SEM states that it does not impact on the destination’s brand (β = 0.146, n.s.), so hypothesis H4a was refuted. On the other hand, based on the results, the perceived quality of a restaurant positively impacts on the overall tourists’ satisfaction (β = 0.355, *p* < 0.05), which allows us to confirm the H4b hypothesis. Thus, this observation was in line with most of the studies reported [43]. In fact, the technical and functional quality of a restaurant has a positive and significant impact on the tourists’ post-trip evaluation and, consequently, on their satisfaction rate, which confirms the outcomes of Björk and Kauppinen-Räisänen [44]’s research. In contrast, the studies of Karim and Chi [98] have not found that a positive evaluation of a restaurant’s quality and the dining experience contribute for strengthening the destination’s brand. Nevertheless, the participants of the current research did not find that the value of a restaurant effected the Ukrainian brand which, respectively, did not confirm this correlation.

The findings indicated that the gastronomic experience was a significant factor affecting the perception of a destination’s brand (β = 0.516, *p* < 0.000) and the tourists’ satisfaction (β = 0.355, *p* =0.022), and this relationship confirmed hypotheses H5a and H2b, respectively. This finding was consistent with other studies in the tourism and hospitality literature (reference). Thus, the obtained results confirmed the findings of Silkes et al. [98], who argued that a positive gastronomic experience obtained in the destination contributes not only to creating a memorable experience but also to enhancing the overall brand image of the destination. This outcome is also in line with Williams et al. [59] who considered the gastronomic experience to be a crucial determinant of a destination’s brand image. From this prospect, crafting a unique and authentic gastronomy experience is a vital condition for creating a prominent and recognizable brand image [58]. Therefore, this result will apply to shaping a destination’s brand within the Ukrainian prospect too. Taking into consideration satisfaction, the observation has shown a positive and significant relationship with the gastronomy experience, which is consistent with some previous findings. As Getz [64] and Ignatov and Smith [99] argued, the gastronomy experience is an exceptional source of the tourists’ satisfaction, and the same outcome was confirmed. Moreover, some other researchers have previously pointed out that the level of satisfaction varies and is related to the food experience which the tourists receive in the destination [65]. More precisely, Guan and Jones [46] and Berbel-Pineda et al. [81] pointed out that travelers who highly rate the quality of their gastronomy experience tend to express satisfaction with the whole trip.

However, the study also found that the perception of the quality of the cuisine was not significantly related to the destination’s brand (β = 0.046, n.s.) and the tourists’ satisfaction (β = 0.007, n.s.). Hence, given these results, the hypotheses H6a and H6b were not confirmed. Despite the findings from the previous studies which conveyed that the quality of food plays a prominent role in the tourists’ satisfaction as well as helps to strengthen a destination’s brand image, the respondents of the survey who have travelled to Ukraine did not find this correlation. This fact might be explained by the tourists’ evaluation of Ukrainian cuisine through what they tried in a restaurant and via different activities. Thus, due to the fact that a limited number of restaurants present authentic Ukrainian food, they were not able to distinguish correctly traditional cuisine. From this prospect, distinguishing the identity of Ukrainian cuisine and presenting what is real and authentic food requires more attention from tourism policy makers. For the broader spreading of the awareness of gastronomy traditions and a better communication, more marketing materials such as books, brochures, magazines, blogs on social media, vlogs, short films, and a variety of welcoming events should be taken into consideration, as suggested by Pereira et al. [100].

Moreover, as far as cuisine was evaluated by multiple items, such as its quality, variety, exotic cooking methods, and taste, the outcome was different than previous studies by Karim and Chi [98]. Another argument is that Ukraine has experienced Soviet occupation, which resulted in issues with identifications of the authenticity of traditional food. During this long period, authentic and unique cuisine were purposefully destroyed and assimilated with Soviet culture. However, due to the independence of Ukraine, a rapidly developing restaurant industry and artisan food production in Ukraine have attracted international food tourists and foreign investors. Nonetheless, all too often some of the most popular dishes known all over the world are not seen as part of Ukrainian cooking, national culinary traditions, and Ukrainian cuisine as a brand. Moreover, the country continues to analyze and rethink their culinary traditions and rediscovers the abundance of flavor.

Apart from the PLS- SEM results of the structural model, an IPMA analysis was also conducted for a further exploration of the results and to analyze the differences regarding the importance of each variable in the outcomes.

Hence, the contribution of using the importance–performance matrix analysis (see Figure 2) is that it distinguishes the vital factors which should be given special attention for providing a memorable gastronomic experience. Speaking more precisely, we could observe from Figure 2 that the gastronomy experience is significantly influenced by past experience (0.661), while the gastronomy experience has an impact on the guests’ satisfaction (0.420) and a destination’s brand image (0.507), respectively. At the same time, this resulted in an IPMA which presented that past experience influences a tourist’s perception of restaurants (0.557). Interestingly, the results have confirmed the significant importance of prior knowledge with respect to the perception of food activities (0.602).

## 5. Conclusions

### 5.1. Theoretical Contributions

Gastronomy tourism, and the gastronomy experience, have received increasing attention in recent decades. There is no doubt that the relationship between gastronomy, the tourist, and the destination within the tourism framework are dynamic and complex [17]. Previous studies which attempted to examine the influence of gastronomy experiences on the tourists’ overall satisfaction and a destination’s brand were completed in a limited context. Specifically, there is a lack of studies concerning the importance of the gastronomy experience for the Ukrainian tourism industry. Thus, the value of gastronomy in the destination could be misleadingly underestimated. The study was designed to provide a complex approach of the gastronomic experience, considering its correlation with the tourists’ satisfaction and a destination’s brand, encountering other factors as well as the highlight the importance of developing gastronomic tourism in Ukraine. Our reasoning was to investigate how the perception of cuisine, restaurant culture, food activities, prior knowledge, and past experience impact the gastronomy experience and how it further influences the tourists’ satisfaction and destination brand. That was done through creating an SEM model. Moreover, as far as the paper aimed to clarify the importance of gastronomy for Ukraine, the significance of the popularization of Ukrainian gastronomy as a tool for the prosperity of the national economy in the post-war period was introduced.

Consequently, from the theoretical prospects, the study has enriched the body of the literature regarding the gastronomy experience. The study contributed to the theoretical understanding of the key factors that influence memorable gastronomic experiences and the relationship between the experience of food and its role in the tourists’ satisfaction and the perceived brand of a destination [70]. The current findings identified a linkage between the perception of the food activities, cuisine, restaurants, prior knowledge, and past experience. Therefore, the obtained results with after statistical testing led to the creation of an SEM that shows a complex correlation between the gastronomy experience as a central element and multiple items that have a direct impact on it or are perceived within the post-evaluation and results of the trip.

The findings of the study contribute to the existing literature by suggesting that past experience and prior knowledge have a positive influence on the gastronomy experience of the tourists in the destination. Furthermore, a significant effect of prior knowledge on the perceived quality of the cuisine and the food activities in the destination of choice was found. The resulting conceptual model allows us to state that the gastronomy experience at the destination is important not only for gaining satisfaction with the quality of the food and dining service, but it is also crucial for the tourists’ overall satisfaction with the trip. Therefore, the results brought the light to the concept that the tourists’ satisfaction is shaped by the multiple gastronomy features of the destination instead of a single prominent factor. Another vital aspect of the findings is that restaurants play an important role in the overall satisfaction of the tourists, which confirmed the findings of previous studies. Although, restaurants from the Ukrainian prospect did not show a significant influence. Taking into consideration the destination’s brand image, testing the model allowed us to conclude that the gastronomy experience is the crucial factor in enriching the destination’s brand. There was found to be a consistent impact of past experience on the gastronomic experience and on the overall satisfaction with the trip and strengthening the destination’s brand, respectively. 

It is worth mentioning that a lot of the previous studies in the academic literature paid attention to the potential attitudes of tourists toward the destination’s image and primarily on the pre-travel stage [70]. Moreover, Ryu and Jang [43] postulated that an impressive food experience helps to enhance the brand image of the country, increase the level of the tourists’ satisfaction, as well as encourage them to return. Thus, the current study is consistent with the previous findings [70] which empirically and theoretically confirm that the gastronomy experience affects the destination. Furthermore, in line with previous research, the current findings reveal that the effective gastronomy experience provided in Ukraine might be a powerful tool for the further promotion of tourism and strengthening the brand image of the country.

### 5.2. Practical Implications

The research provides not only a theoretical contribution but also practical implications. Those findings contribute to the understanding of the relationship between the gastronomic experience and the variables that compose the model, analyzing how, through the gastronomy experience, the tourists’ satisfaction and building a distinguishable brand for the destination could be achieved. Moreover, other factors that shape the gastronomy experience in the destination of the choice, such as prior knowledge, restaurants, past experience, cuisine, and food activities, were considered to show its correlation with other items as they are important for predicting the tourists’ behavior and evaluation of the trip. 

The obtained SEM revealed that the dimension of past experience had a great impact on the people’s perception of the cuisine’s quality, restaurant’s quality, as well as on the overall gastronomic experience. This result reflects for the destination’s marketers and policymakers that a real-time food experience could be perceived as a precursor to creating a constructive tourist past experience and, as a result, deliver a positive perception of the quality of the local cuisine and dining places, as well as increase the value of the current food experience in the destination. Therefore, the prior knowledge dimension has shown to have a considerable influence on the perception of the food activities, cuisine’s quality, and the gastronomy experience, accordingly. Thus, practitioners of the tourism industry are advised to supply tourists with proper information feeds regarding food specialties that should be experienced to gain a memorable gastronomy experience, as well as activities and locations where local cuisine might be savored. As a result, accessible and relative food-related information will lead to a higher level of satisfaction for tourists with not only the gastronomy experience but with the overall trip.

A variety of tourism stakeholders should work closely to provide and maintain high-quality food and service. Considering local restaurants, it was found that the experiences gained in different [70] dining places greatly influence people’s satisfaction with the trip. Destination restaurants have been distinguished as a unique condition to develop an attachment to a place in visitors. Consequently, practitioners of the tourism industry could work more closely with local restaurants to create memorable tourism experiences and project a further attractive brand image for a destination. Furthermore, marketers, policymakers, and other stakeholders can utilize restaurants as their flagship experience. Based on the obtained result, practitioners should bear in mind restaurants as strategic selling points where, through a unique gastronomy experience, a destination’s culture will be effectively shown. Additionally, validated by these findings, service providers should consider the linkage between restaurants and the destination itself in their ability to perhaps showcase local gastronomy that, respectively, influences the tourists’ further satisfaction with their overall trip. Destination management should emphasize the national food image to effectively promote destination tourism and the country’s identity [74].

### 5.3. Limitations and Future Research

The study is an initial attempt to find the relationship between the gastronomy experience in a destination and the factors that may influence it by building a structural equation model. Even though we shed a light on the topic, the study has some limitations which bring fruitful avenues for future research. First, the current study has a methodological limitation due to the purposive sampling procedure which limits the generalization of the results. Additionally, some bias can be found in the sample age average, showing many young people, who tend to have a low income to spend on food. It would be useful to apply to a larger number of respondents and to collect answers not only via virtual tools. The researchers could also apply to visitors in different types of servicescapes such as restaurants, food markets, fairs, restaurants, food festivals, and other events to obtain deeper insides. There is an evident fact that different environments can influence and shape the tourists’ evaluation which might be used for the further development of a destination. Additionally, taking into the consideration the quantitative method of the current study, it has been suggested to use a qualitative approach to gain an in-depth understanding of the gastronomy experience.

Second, given the geographical area of the research in Ukraine, however, it would be interesting to collect answers about tourist experiences in other gastronomy destinations, ones which are either more or less popular. Thus, a comparison could be made between the different tourist experiences obtained in the destination and the different levels of gastronomy tourism. Moreover, we consider that it might be useful to collect answers from respondents with a variety of cultural backgrounds to have an efficient overview of gastronomy experiences in the global context. Third, the study was mainly focused on demand, which makes it difficult to convey the results to other subjects of gastronomy activity such as stakeholders, local communities, or businesses. 

Given the current lack of studies concentrated on gastronomy experiences holistically and systematically, the structural model developed is believed to contribute to the body of knowledge in the tourism field as well as deliver the theoretical groundwork for further research and to finding a practical solution in the tourism sector.

## Figures and Tables

**Figure 1 foods-12-00315-f001:**
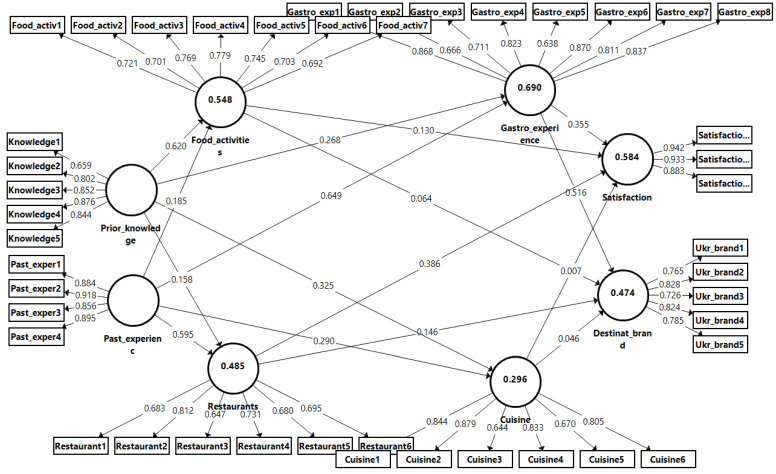
Structural model. Note: the circles represent the constructs or variables; the rectangles represent the items of the construct’s measurements, with the associated values for the standardized factor loadings; and the arrows represent the tested relationships between the constructs, with the beta values.

**Figure 2 foods-12-00315-f002:**
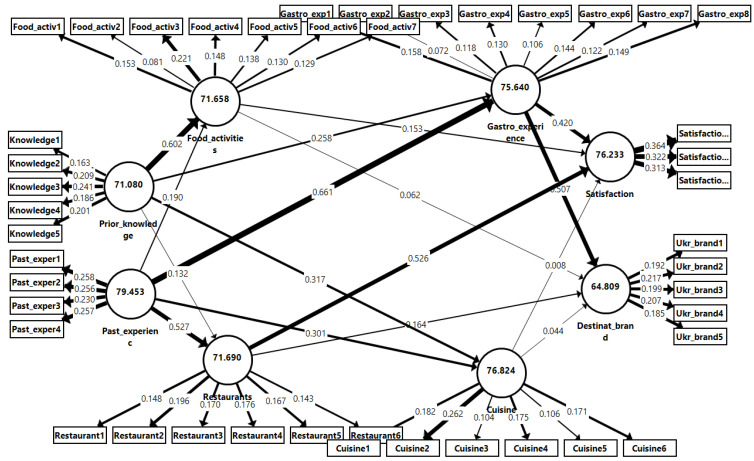
Structural model using IPMA.

**Table 1 foods-12-00315-t001:** Measures.

Construct	Items
**Knowledge** Leong, Q.-L., Ab Karim, S., Awang, K.W. and Abu Bakar, A.Z. [7].	Read about local food prior to travel.
	Aware of local eating customs.
	Knowledgeable about local food.
	Informed about popular local food.
	Informed of the location of popular local food.
**Past experience** Leong, Q.-L., Ab Karim, S., Awang, K.W. and Abu Bakar, A.Z. [7].	Enjoyable.
	Good quality service.
	Learnt about the local food culture.
	Enhanced travel experience.
**Destination brand** José A. Folgado-Fernández, José M. Hernández-Mogollón, and Paulo Duarte [79]	Good infrastructures.
	Well trained/good workmanship.
	Good living and working conditions.
	Communication of an appealing vision of the country.
	Attractive image.
	I will recommend this destination to family and friends.
**Food/Cuisine** Shahrim Ab Karim and Christina Geng-Qing Chi [80]	Offers a variety of foods.
	Offers good quality food.
	Offers regionally produced food products.
	Offers attractive food presentation.
	Offers exotic cooking methods.
	Offers delicious food.
**Dining/Restaurant** Shahrim Ab Karim and Christina Geng-Qing Chi [80]	Offers reasonable price for dining out.
	Offers many attractive restaurants.
	Offers easy access to restaurants.
	Offers varieties of specialty restaurants.
	Offers friendly service personnel.
	Offers restaurants menus in English.
**Food-related tourism activities** Shahrim Ab Karim and Christina Geng-Qing Chi [80]	Offers food and wine regions.
	Offers package tours related to food and wine.
	Offers a unique cultural experience.
	Offers an opportunity to visit street markets.
	Offers unique street food vendors.
	Offers various food activities, e.g., cooking classes and farm visits.
	Offers much literature on food and tourism.
**Gastronomic experience** Moral-Cuadra, S., Acero de la Cruz, R., Rueda López, R. and Salinas Cuadrado, E. [78].	Quality of the dishes.
	Price.
	Installation.
	Atmosphere of the establishment.
	Innovation and new flavors of the dishes.
	Service and hospitality.
	Experience with traditional gastronomy.
	Offers genuine gastronomic products.
**Satisfaction** Berbel-Pineda, J.M., Palacios-Florencio, B., Ramírez-Hurtado, J.M. and Santos-Roldán, L. [33].	How important is gastronomy for you as a destination choice when travelling?
	How important are gastronomic experiences for you when you choose a destination for your trip?
	How important is gastronomy for you in relation with the satisfaction of your trip?
	My level of satisfaction with the gastronomy has been significant.

**Table 2 foods-12-00315-t002:** Respondents profile.

Variable		Percentage	Variable		Percentage
Gender	Female	76.4%	Country of origin	Ukraine	56.4%
	Male	23.6%		Other	43.6%
Age	18–29	76.4%	Gastronomic experience during the trip	Yes	95%
	30–39	20%		No	5%
	40–49	1%			
	50–59	1%			
Professional activity	Student	36.4%	Education	High school	17.5%
	Full-time/part-time job	45.5%		Associate degree	21.3%
	Self-employed/freelance	16.4%		Bachelor’s degree	43%
				Master’s degree	18.2%
	Unemployed	1.8%			

**Table 3 foods-12-00315-t003:** Composite reliability, average variance extracted, correlations, and discriminant validity checks.

Latent Variables	α	CR	AVE	1	2	3	4	5	6	7	8
(1) Cuisine	0.874	0.904	0.615	**0.784**	0.437	0.257	0.491	0.493	0.533	0.635	0.433
(2) Destination_brand	0.845	0.890	0.619	0.384	**0.787**	0.538	0.752	0.689	0.658	0.645	0.688
(3) Food_activities	0.856	0.889	0.534	0.217	0.477	**0.731**	0.718	0.593	0.829	0.580	0.586
(4) Gastro_experience	0.908	0.926	0.613	0.469	0.673	0.643	**0.783**	0.776	0.701	0.743	0.744
(5) Past_experience	0.911	0.937	0.789	0.473	0.607	0.534	0.801	**0.888**	0.626	0.787	0.663
(6) Prior_knowledge	0.866	0.904	0.656	0.488	0.565	0.724	0.634	0.563	**0.810**	0.583	0.619
(7) Restaurants	0.802	0.858	0.504	0.566	0.537	0.489	0.646	0.684	0.493	**0.710**	0.802
(8) Satisfaction	0.908	0.942	0.845	0.420	0.605	0.549	0.692	0.611	0.557	0.683	**0.919**

**Note**: α—Cronbach alpha; CR—composite reliability; AVE—average variance extracted. Bolded numbers are the square roots of AVE. Below the diagonal elements are the correlations between the constructs. Above the diagonal elements are the HTMT ratios.

**Table 4 foods-12-00315-t004:** Structural model assessment.

Path	Path Coefficient	Standard Errors	*t* Statistics	*p* Values
Past_experience ^®^ Cuisine	0.290	0.133	2.185	0.029
Past_experience ^®^ Food_activities	0.185	0.130	1.425	0.155
Past_experience ^®^ Gastro_experience	0.649	0.102	6.370	0.000
Past_experience ^®^ Restaurants	0.595	0.123	4.829	0.000
Prior_knowledge ^®^ Cuisine	0.325	0.132	2.464	0.014
Prior_knowledge ^®^ Food_activities	0.620	0.101	6.147	0.000
Prior_knowledge ^®^ Gastro_experience	0.268	0.100	2.690	0.007
Prior_knowledge ^®^ Restaurants	0.158	0.119	1.330	0.184
Food_activities ^®^ Destination_brand	0.064	0.123	0.522	0.602
Food_activities ^®^ Satisfaction	0.130	0.121	1.080	0.281
Restaurants ^®^ Destination_brand	0.146	0.165	0.883	0.378
Restaurants ^®^ Satisfaction	0.386	0.181	2.134	0.033
Gastro_experience ^®^ Destination_brand	0.516	0.139	3.712	0.000
Gastro_experience ^®^ Satisfaction	0.355	0.155	2.295	0.022
Cuisine ^®^ Destination_brand	0.046	0.145	0.317	0.752
Cuisine ^®^ Satisfaction	0.007	0.114	0.062	0.950

## Data Availability

Data are available upon reasonable request to the corresponding author.

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
