# Peer review of "Gastronomic Experience and Consumer Behavior: Analyzing the Influence on Destination Image"

_foods, 2023, doi:10.3390/foods12020315_

Round 1

Reviewer 1 Report

An interesting article, supported by current knowledge on the subject, although the language used by the authors could be more scientific, sometimes too philosophical.

I appreciate the information collected by the authors on Ukraine, focused on its gastronomic potential. However, due to the difficult situation in Ukraine, it is difficult to consider the possibility of tourists traveling for culinary purposes.

I am concerned about the small number of people participating in the study. Can the presented analyzes be considered authoritative, since half of the respondents are people of Ukrainian origin? Even if these people do not live in Ukraine, they certainly had some contact with culture, dishes, flavors and this certainly influenced their answers. Other comments:

Tables and figures have different numbers than those referred to in the text, which causes difficulties in the reception of the work. Please, check the numbering carefully and make appropriate corrections in the tables/drawings and the text of the thesis.

L94 and elsewhere. Please change word 'chapter' to word 'section'. ‘Section’ is preferred for articles, ‘chapter’ for books.

L471-472. Over what period was the questionnaire conducted? Was its coverage hindered by the situation in Ukraine? For example, a smaller number of followers on social profiles.

L493-510. It would be worth attaching the questionnaire as an additional "supplementary file", which would make the work clearer and facilitate the interpretation of the results.

L514-526. How was the statistical analysis of the results performed? And at what significance level?

L529. Please provide more information about the travellers, were they frequent visitors to Ukraine or were they only once?

L533. '30-9'? Correct

Table 2. Country of origin – Ukraine 56.4%. How to interpret the participation of Ukrainians in the questionnaire? Have these people been living outside Ukraine since birth?

L595-596. On what basis is this statement? Please provide literature on this fact.

L632-634. At what level of significance was the statistical analysis performed? There is no information on this in the methodology of paper.

L668-671. Q2 values? Where did these values come from?

Table 3. The number '8' is missing at the end of the first row in the table.

Figure 1. What are the results of individual indexes for given categories, e.g. Restaurant1….Restaurant6? This should be better described in the methodology.

L791-798.  IPMA analysis was not included in the methodology.

L913-915. ‘…survey was also conducted mostly among first-time visitors who are not very familiar with the destination and the area of Ukraine.’ On what basis is this statement? There is no data on this subject in the content of the paper. How does this relate to people of Ukrainian origin in the survey? More than half of the respondents are people of Ukrainian origin.

Linguistic errors, example: country's` (L74), dis (L83)…

Author Response

An interesting article, supported by current knowledge on the subject, although the language used by the authors could be more scientific, sometimes too philosophical.

R: Thank you for the supportive comment and for the recommendations to improve our manuscript. We tried to carefully address all the topics.

I appreciate the information collected by the authors on Ukraine, focused on its gastronomic potential. However, due to the difficult situation in Ukraine, it is difficult to consider the possibility of tourists traveling for culinary purposes.

R: Yes, we agree with the reviewer that this are difficult times for Ukraine. We strongly hope this article will be useful when peace comes. Tourism will certainly help economic growth.

I am concerned about the small number of people participating in the study. Can the presented analyzes be considered authoritative, since half of the respondents are people of Ukrainian origin? Even if these people do not live in Ukraine, they certainly had some contact with culture, dishes, flavors and this certainly influenced their answers. Other comments:

R: Thank you for these comments. As detailed below we addressed your recommendations regarding the small sample and the inclusion of domestic tourists in the sample. Further detail is presented in the topics #6 and #11.

Tables and figures have different numbers than those referred to in the text, which causes difficulties in the reception of the work. Please, check the numbering carefully and make appropriate corrections in the tables/drawings and the text of the thesis.

R: We are thankful for the reviewer for pointing this mistake. We revised the text references to the tables and table numbering for consistency. We believe it is all correct now.

L94 and elsewhere. Please change word 'chapter' to word 'section'. ‘Section’ is preferred for articles, ‘chapter’ for books.

R: Thank you for pointing this out. The word chapter has been replaced by section.

L471-472. Over what period was the questionnaire conducted? Was its coverage hindered by the situation in Ukraine? For example, a smaller number of followers on social profiles.

R: We are thankful for this comment and request for clarification. In fact, we intended to obtain a larger sample, but due to the beginning of the war at the end of February 2022, data collection was suddenly interrupted. As such, our data was collected before the war, between dec 2021 and the beginning of the war. We added this information to section 3.1. Measures, but we did not add any information regarding the war.

L493-510. It would be worth attaching the questionnaire as an additional "supplementary file", which would make the work clearer and facilitate the interpretation of the results.

  1. We understand the reviewer concern. It is important for other willing to replicate our study (or related) to use a pre-defined questionnaire. Table 1 provides a detailed presentation of the questionnaire items used in the survey. If the reviewer agrees this could substitute the supplementary material avoiding to duplicate information. Otherwise, we will be glad to provide the questionnaire ready to use in a separate file.

L514-526. How was the statistical analysis of the results performed? And at what significance level?

R: We agree that further explanation about the procedures were necessary. Thank you for this recommendation. Accordingly, we expanded section 3.1. to provide additional detail about the quality assessment procedures to assess the constructs and the conceptual model.

L529. Please provide more information about the travellers, were they frequent visitors to Ukraine or were they only once?

R: We agree that this information is important in the context of this study. As such, we added that we surveyed people that visited at least once Ukraine as an inclusion criteria. This information is now included in the beginning of section 3.2.

L533. '30-9'? Correct

R: Thank you for pointing this out. The age reference was corrected to 30-39.

Table 2. Country of origin – Ukraine 56.4%. How to interpret the participation of Ukrainians in the questionnaire? Have these people been living outside Ukraine since birth?

R: This is a very important question. Our goal is to analyze the influence of the variables independently the country of origin. As such, we considered domestic tourists as part of the sample as other studies in gastronomic tourism – we provide the following references to support this choice: Björk, P., & Kauppinen-Räisänen, H. (2014); Ullah, N., Khan, J., Saeed, I., Zada, S., Xin, S., Kang, Z., & Hu, Y. (2022).

Still, we totally agree that this could cause confusion among readers. As such, we extended the sample description to provide this information in the 2nd paragraph of section 3.2.

L595-596. On what basis is this statement? Please provide literature on this fact.

R: Thank you for the recommendation. We now added a citation to support the affirmation.

L632-634. At what level of significance was the statistical analysis performed? There is no information on this in the methodology of paper.

R: Thank you for alerting to this important issue. Due to the fact that the total number of members of the population is indetermined, to establish a sampling frame is impossible. As such, we followed non-probability sampling procedure, more specifically, a purposive sampling technique was adopted. This methodological option is now added to the sampling procedures in section 3.2.. We also recognize the limitation associated to this procedure in the conclusions. This information can be consulted in the 2nd paragraph of the limitations.

L668-671. Q2 values? Where did these values come from?

R: We agree that this topic was not clear. We reformulated the sentence to provide the procedures (blindfolding) to obtain this model quality indicator.

Table 3. The number '8' is missing at the end of the first row in the table.

R: Thank you for the sharp attention. The number 8 is now included in the first row.

Figure 1. What are the results of individual indexes for given categories, e.g. Restaurant1….Restaurant6? This should be better described in the methodology.

R: Thank you for pointing this out. We now present an explanation for this figure as an output of PLS-SEM. This explanation is provided a note for figure 1 explaining that the circles represent the constructs or variables; the rectangles represent the items of the construct’s measurements, with the associated values for the standardized factor loadings; the arrows represent the tested relationships between the constructs.

L791-798.  IPMA analysis was not included in the methodology.

R: We agree with the reviewer. The methodology should describe all the procedures. As such we now provide information regarding IPMA procedure ate the end of section 3.1. For coherence, we also reformulated the sentence in the former lines L791-798 as a consequence of this change.

L913-915. ‘…survey was also conducted mostly among first-time visitors who are not very familiar with the destination and the area of Ukraine.’ On what basis is this statement? There is no data on this subject in the content of the paper. How does this relate to people of Ukrainian origin in the survey? More than half of the respondents are people of Ukrainian origin.

R: Thank you for pointing this out. We agree that the identified sentence is incorrect due to our sampling frame. What we meant is related to the sampling procedure. The sentence was revised accordingly.

Linguistic errors, example: country's` (L74), dis (L83)…

R: Thank you for the recommendation. These errors were revised. We also conducted a proofread of the entire document to identify and correct other situations.

Reviewer 2 Report

Thank you for giving me the opportunity to read and reflect on your paper, I think this is well structured and well written. It is a good paper.However in my opinion, some important details should be emphasized in order to improve the quality of the research:

Firstly, I am concerned that the selection of the sample was mostly snowball effect, as I believe it lacks scientific robustness, could you justify this with previous references?

Most of the sample, according to section 3.2 Sample, is made up of people between 18 - 29 years old, 76.4% to be precise, I consider that this age range does not have the purchasing power to travel for gastronomic tourism, as usually the restaurants that are a reference in this sector, the gastronomic experience usually has a high cost, could you justify this?

Finally, could you justify that a sample of this size is relevant to be conclusive in a research?

Author Response

Thank you for giving me the opportunity to read and reflect on your paper, I think this is well structured and well written. It is a good paper.However in my opinion, some important details should be emphasized in order to improve the quality of the research:

R: Thank you for the supportive comment and for the recommendations to improve our manuscript. We tried to carefully address all the topics.

Firstly, I am concerned that the selection of the sample was mostly snowball effect, as I believe it lacks scientific robustness, could you justify this with previous references?

R: We agree with the reviewer that snowball sampling has certain shortcomings. As such we, totally understand the reviewer’s concern. As recommended by the reviewer we added references from previous research in tourism adopting this technique, as well as the reasons those authors adopted this technique. This can be consulted in the first paragraph of section 3.2.

Most of the sample, according to section 3.2 Sample, is made up of people between 18 - 29 years old, 76.4% to be precise, I consider that this age range does not have the purchasing power to travel for gastronomic tourism, as usually the restaurants that are a reference in this sector, the gastronomic experience usually has a high cost, could you justify this?

R: Thank you for pointing this out. This is a very important and sharp question. We can partially justify this by considering that gastronomy is part of the tourist travelling experience. For this perspective we added additional justification and the supporting reference from Björk and Kauppinen-Räisänen (2014) on section 3.2.. Of course, this only explains partially. As such we now recognize this bias in the limitations at the end of the document.

Finally, could you justify that a sample of this size is relevant to be conclusive in a research?

R: Thank you for alerting to this important issue. Due to the fact that the total number of members of the population is indetermined, to establish a sampling frame is impossible. As such, we followed non-probability sampling procedure, more specifically, a purposive sampling technique was adopted. This methodological option is now added to the sampling procedures in section 3.2.. We also recognize the limitation associated to this procedure in the conclusions. This information can be consulted in the 2nd paragraph of the limitations.